# Early Detection of Refractive Errors by Photorefraction at School Age

**DOI:** 10.3390/ijerph192315880

**Published:** 2022-11-29

**Authors:** Marta Alvarez, Clara Benedi-Garcia, Pablo Concepcion-Grande, Paulina Dotor, Amelia Gonzalez, Eva Chamorro, Jose Miguel Cleva

**Affiliations:** Indizen Optical Technologies S.L., 28002 Madrid, Spain

**Keywords:** amblyopia, myopia, high myopia, visual screening, refractive errors

## Abstract

Early detection and treatment of refractive defects during school age are essential to avoid irreversible future vision loss and potential school problems. Previously, vision screening of preschool children used methods based on subjective visual acuity; however, technologies such as photorefraction have promoted the detection of refractive errors quickly and easily. In this study, 1347 children from 10 schools in Madrid aged 4 to 12 years participated in a program of early detection of visual problems, which consisted of visual screening composed of anamnesis and photorefraction with a PlusOptix A12R. The prevalence of refractive errors was analyzed in terms of spherical equivalent, cylinder and its orientation, and potential cases of development of high myopia or amblyopia. Hyperopia predominates in the early years, but the number of myopic subjects is higher than that of hyperopic subjects from the age of ten onwards. At all ages, the predominant orientation of astigmatism was with-the-rule. On average, 80% of the myopic subjects were uncorrected. Potential high myopia increased with age, from 4 to 21% of the measured population. Potential amblyopia cases decreased across age groups, from 19 to 13.7%. There is a need to raise awareness of the importance of vision screening at school age to address vision problems.

## 1. Introduction

The World Health Organization (WHO) estimates that 19 million children have visual impairment. In addition, it is known that the main cause of visual impairment worldwide at school age is uncorrected refractive errors [1]. Undetected refractive errors can affect school learning and performance and cause ocular discomfort, attention deficit, avoidance of homework, and the development of a negative association between visual discomfort and educational activities [2].

Myopia has been recognized as a significant public health problem worldwide [3] as its progression to high myopia carries high risks of irreversible blindness complications, caused by glaucoma, retinal detachment, and myopic maculopathy [4]. In the last decades, earlier onset of myopia has been observed in East and Southeast Asia [5], and it is widely recognized that the earlier the age of onset, the higher the percentage of the population with high myopia [6]. It is, therefore, of great interest to implement myopia prevention strategies to reduce or delay the early onset of myopia [7].

Among the visual disorders that can affect the normal development of vision in childhood, apart from refractive errors, are strabismus and anisometropia. These alterations can lead to the development of amblyopia, which is one of the most common causes of unilateral or bilateral vision loss without ocular pathology [8], with a worldwide prevalence ranging from 0.2% to 6.2% in children [9]. In some cases, treatment of refractive defects with glasses may solve refractive amblyopia.

Early detection and intervention of refractive errors and amblyopia are essential to reduce vision loss in childhood and improve quality of life in adulthood. In many cases, childhood vision disorders are asymptomatic and can often go undetected and untreated. To avoid potential school problems resulting from the non-correction of refractive errors, vision screening programs are needed in schools within the developmental period of the child [10,11]. The goal of these visual screening programs is to detect refractive errors or visual problems quickly, not to diagnose or quantify diopters. After detection, they must be checked by eye care professionals with a detailed subjective refraction.

For these visual screenings, there are needed instruments that can be used in screening large populations to reliably detect both refractive errors and other amblyogenic risk factors. Several studies have shown an overall agreement between photorefraction and cyclo-autorefraction ranging from 81.18% to 94.12%, using both amblyogenic and non-amblyogenic refractive error criteria [12,13]. Fogel-Levin et al. [10] evaluated the accuracy and reliability of PlusOptix by undergoing screening with the PlusOptix device and cycloplegic refraction. Their findings were that the reliability was lower in hyperopic eyes, so they referred slightly hyperopic children in the PlusoptiX screening for cycloplegic refraction. On the other hand, Paff et al. [14] examined the Retinomax K-plus2 with cycloplegia and showed a significant improvement in the diagnosis of hyperopic eyes in comparison with PlusOptix photorefractometer. In contrast, Fogel-Levin et al. found that accuracy was very good in the myopia, astigmatism, and anisometropia groups and, therefore, considered the PlusOptix device a fast, simple, and secure tool for the detection of refractive errors.

In addition, the measurement of refraction with the PlusOptix device has more advantages over other methods such as the following: the simultaneous measurements of both eyes help detect anisometropia, the noncycloplegic refractions may be closer to real-life refractions, the technique has been shown to detect astigmatism reliably if the refractive errors are in the linear range of measurement, and myopia of more than one diopter is reliably measured [15]. Therefore, photorefraction could be considered as an alternative to widespread school screening programs [16,17,18].

Photorefraction is a system inspired by the Brückner test and is an easy method to detect refractive problems, strabismus, and media opacities [19]. Over the years, these devices have been considerably improved and now measure refraction using the principle of transillumination or backlighting. Photorefraction avoids patient glare by using infrared light, allowing the pupil size to remain large even without dilation. These non-contact devices have the advantage that subjects do not have to rest their heads on a chin rest as would be the case with traditional autorefractometers. Furthermore, measurements can be taken at working distances of 1 m making the device more suitable for examining small and uncooperative children [18,20]. The aim of the study is to implement a program for early detection of refractive errors in schools with a photorefractometer and to determine the prevalence of refractive errors measured, estimating the percentage of potential myopic, myopic, and amblyopic children in a pediatric population of several schools in the city of Madrid (Spain).

## 2. Methods

### 2.1. Patients

This is a cross-sectional study in which refractive data were collected from a population of children aged between 4 and 12 years (mean ± SD, 7.98 ± 2.29). Through the AMIRES association (Magna Myopia with Retinopathies Association), 10 schools in Madrid were contacted. All students within the age range were invited to participate in a program of early detection of visual problems not previously detected at school. Optometrists measured the refraction of a total of 1347 participants.

The parents or legal guardians of the participants were informed of the tests by the schools and of the study following the tenets of the Declaration of Helsinki. The data used in this study were collected during the recruitment phase of a larger study approved by the Ethics Committee of the Hospital Clínico San Carlos and registered in the U.S. National Library of Medicine under the identifier NCT05250206.

### 2.2. Data Collection

The test consisted of an anamnesis and a quick assessment with an autorefractometer. In the anamnesis, participants were asked their sex, age, and whether they wore glasses. Photorefraction was performed with the PlusOptix A12R instrument (Plusoptix GmbH, Nuremberg, Germany) [10,21,22] simultaneously in both eyes, at one meter, and under mesopic lighting conditions, which facilitated pupil detection and relaxed accommodation. During the measurements, these devices produce noises and flashing lights to help children maintain a steady gaze [16]. The optometrist checked the repeatability of the instrument by taking two consecutive measurements per child. If both differed by more than 0.50 D, they were measured again. Measurement time was less than 30 s, and each participant spent a total of 5 min with the examiner.

### 2.3. Analysis

The refractive error of the subjects was studied according to the spherical equivalent, the cylinder and its orientation, the potential probability of developing high myopia in the future, and the risk of suffering amblyopia due to their refraction. For the analysis, the thresholds described in Table 1 were used. The results are shown in terms of percentage of subjects. Gender differences were analyzed with the Wilcoxon test, and statistical significance was set at *p*-value < 0.05.

## 3. Results

The optometric data of 1347 children were analyzed. The sample consisted of 653 (48.47%) girls and 694 (51.53%) boys, and the distribution among age groups was equal: 30% of the sample was between 4 and 6 years old; 39%, between 7 and 9; and 31%, between 10 and 12.

Figure 1a shows the average spherical error by group of age with the standard deviations, as well as the range of values expected by age. Figure 1a shows that the percentage of subjects with potential refractive errors changes between 20% and 30%, depending on age. Figure 1a shows that for all subjects in whom ametropia was found, on average, 80% were uncorrected. In addition, it was observed that the proportion of myopic versus hyperopic subjects increases with age, with the number of myopic subjects being greater than hyperopic subjects from the age of ten years onwards (Figure 1c). Regarding gender differences, Wilcoxon test was run to compare the medians of the two samples, males and females, with no statistically significant differences between them (*p*-value = 0.74).

On average, 10.5 ± 3.8% of the subjects presented a cylinder higher than 1.5 D, and there is no correlation with age. Regarding the angle of the astigmatism, with-the-rule astigmatism (180° ± 30°) is the most predominant in the measured sample, appearing in 82.73% of astigmatic subjects, compared to 15.83% of against-the-rule astigmatic subjects (90° ± 30°) and 1.44% with oblique astigmatism. Again, Wilcoxon test showed that there were not statistically significant differences between genders (*p*-value = 0.18) for the angle of astigmatism.

Figure 2a shows the percentage of myopic subjects, future myopic subjects, and subjects at risk of developing high myopia according to the screening data. It can be seen that the percentage of myopic subjects increases with age (9% between 4 and 6 years of age, 14% between 7 and 9 years of age, and 23% between 10 and 12 years of age).

In addition, between 60% and 70% of the subjects meet the criteria for developing myopia in the future, with no dependence on age and with a slight decrease at 12 years old (52.38%). The percentage of subjects who could potentially develop high myopia in the future, follows a similar behavior to the percentage of myopes, with an incidence of 4% between 4 and 6 years of age, 6% between 7 and 9 years of age, and 21% between 10 and 12 years of age.

Figure 2b shows the percentage of subjects with refractive errors, potentially amblyopic subjects due to their refraction, and undetected subjects. The percentage of subjects with refractive errors includes both genetic and environmental components, and it is well known that the incidence errors are between 20% and 40% in the studied range of age. However, the risk of amblyopia due to subjects’ refractive errors decreases slightly over the groups of age. It was found that 19% of subjects between the ages of 4 and 6 years are potentially amblyopic, 17.5% in subjects between 7 and 9 years, and 13.7% in subjects between 10 and 12 years.

As a parameter of potentially amblyopic subjects who have not been identified, the number of uncorrected subjects was studied. Following this premise, it was found that 68% of the potentially amblyopic subjects had not yet been identified. Again, the number of uncorrected subjects was found to decrease with increasing age (77% between 4 and 6 years, 67% between 7 and 9 years, and 56% between 10 and 12 years).

## 4. Discussion

A visual screening was carried out on 1347 children with a photorefractometer. Photoscreening is a useful approach to narrow optometric error in a large population. However, results must always be confirmed with a refraction carried out by an optometrist or ophthalmologist with the aim of proving a diagnosis. Although photorefraction has been previously validated [18] and is a recommended technology for vision screenings [26], one of the limitations of the study was the absence of a cycloplegic to paralyze accommodation, which could lead to under-representation of hyperopia [27].

In the case of our data, hyperopia was the most common refractive error (Figure 1c), and while age increases, myopia becomes the predominant ametropia. These trends are consistent with knowledge about the prevalence of refractive errors in childhood [28]. However, the refractions obtained were, on average, 0.7 ± 0.1 D more myopic than would be expected according to McCullough et al. [24]. Nevertheless, the myopia prevalence rates found in our screening are in agreement with the estimations made by the Brien Holden Institute [29]. After conducting a systematic review and a meta-analysis, they estimated the number of people who will be myopic between the years 2000 and 2050. They found that the prevalence of myopia between the ages of 5 and 9 could be around 9.5% and around 20% for ages 10 to 14. On average, 7.27% of the measured subjects who were between the ages of 7 and 14 and 14.35% of the subjects between the ages of 10 and 12 were myopic. Myopia involves both genetic and environmental components, and it is well known that the incidence of myopia is highly dependent on the region of the world. For example, Luong et al. [30] found that East/Southeast Asian children showed noticeably quicker myopia growth over time, and Pan et al. [31] found that the overall rates of high myopia were 4.6% and 11.8% for Chinese subjects. The chosen criteria to establish the risk of developing high myopia in the future was based on a study carried out by Hu at al. [7] in Guangzhou, China, although our population is mainly white. That is because, to our knowledge, no other recent, large-scale study has stratified the risk of high myopia among age in terms diopters on a white population. Therefore, the percentage of subjects who could potentially develop high myopia (Figure 2a) should be carefully considered, and although it makes sense that the curve follows a similar trend to the curve of number of myopes, it could be shifted upward.

When comparing findings of this study with other screenings carried out on similar characteristics population, we found an incidence of myopia lower than 1.9% at 6 years old, similar to Robinson [32] who performed a comparable method with ours (static retinoscopy without cycloplegia). Our data differ to a greater extent from other studies carried out in Europe, which could be explained with the difference on setting, the limits to define myopia and hyperopia, or the methods to collect the date. For example, Jobke et al. [33] found in Germany 5.5% and 6.4% myopia and hyperopia incidence, respectively, for the range of 7–11 years old, compared to our data in which we found 11.4% of myopic and 13.4% of hyperopic children for the same range of age. Furthermore, the refractions were asked to the optometrists of each child. Another screening was performed in Great Britain by Cummings [34] who found an incidence of myopia of 24.4% and 0.6% of hyperopia between 8 and 10 years old, while we found 11.6% and 11.7% of myopes and hyperopes. In this case, the visual exams were performed by the nurses of the schools. In general, our data are in the middle range of other studies, although no one of the studies was carried out in recent years, which could also indicate a variation in the development of the population’s visual profiles. Furthermore, findings regarding cylindrical error seem to match with the known prevalence of astigmatism. For example, more than 80% of the astigmatic subjects presented the cylinder with-the-rule, and a low representation of the astigmatic subjects (1%) had oblique astigmatism. These results are in agreement with other studies on the incidence of astigmatism angle [27].

It was observed that the probability of developing amblyopia (Figure 2b) was higher during the early years, and the probability of developing amblyopia in the range of 4 to 12 years was 16.8% on average. This value means a false referral rate smaller than in other studies of amblyopia risk factors with photoscreening [12,13,14,15,16]. This fact could be explained because our predictions only account for the development of amblyopia due to refractive errors, and other causes such us strabismus or cataracts play a role in amblyopia.

## 5. Conclusions

The prevalence of uncorrected refractive errors is high among children, so an early detection and treatment is key to avoiding irreversible vision loss in the future. Between 60% and 70% of the subjects meet the criteria for developing myopia in the future. In addition, the incidence of subjects who could potentially develop high myopia in the future was 4%, 6%, and 21% for the age groups between 4–6, 7–9, and 10–12 years old, respectively. Potentially amblyopic individuals who have not been identified represented 68% of the total sample. Therefore, it is important to point out that only 16% of the subjects in whom ametropia was detected were corrected with an optical solution. These results show that it is necessary to reinforce school health programs to provide more information and better eye care services to improve this public health problem.

## Figures and Tables

**Figure 1 ijerph-19-15880-f001:**
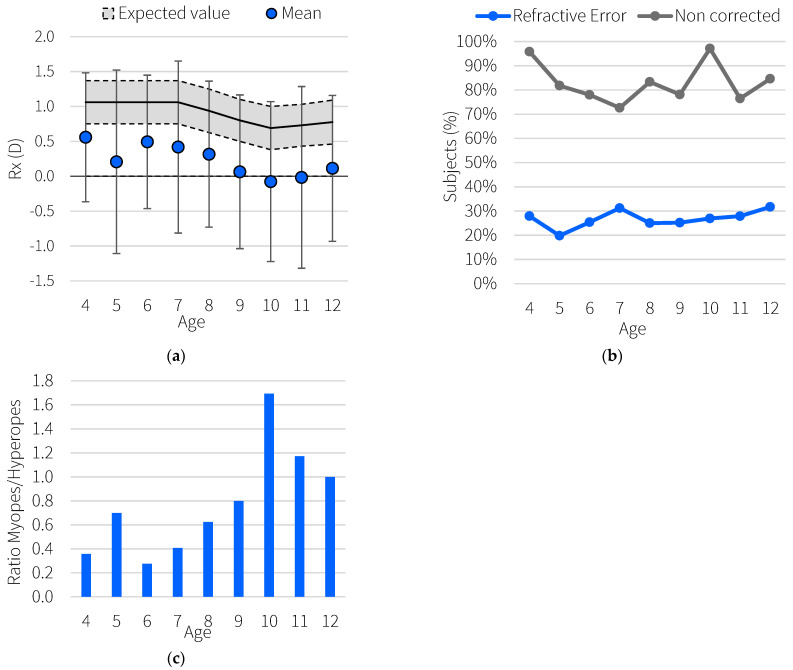
(**a**) Mean value of measured refractions (D) as a function of age. The error bars indicate the standard deviation for each age group, and the shaded area represents the expected values according to McCullough et al. [24] (**b**) In blue, percentage of subjects in whom potential refractive errors were found according to the age. In gray, percentage of subjects with uncorrected refraction. (**c**) Ratio of myopes and hyperopes for each age.

**Figure 2 ijerph-19-15880-f002:**
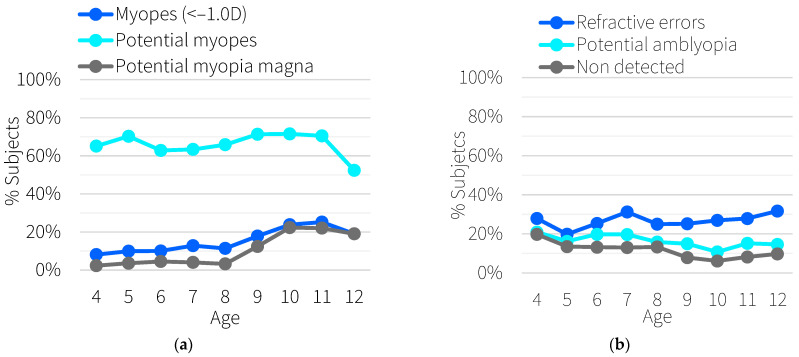
(**a**) Percentage of subjects in the total sample who are myopic (dark blue), at risk of being future myopic (light blue), at risk of having myopia magna (gray). (**b**) Percentage of subjects with potential refractive errors (dark blue), potentially amblyopic (light blue) and uncorrected subjects (gray).

**Table 1 ijerph-19-15880-t001:** Criteria used to classify refractive errors.

Refractive Error	Criteria							
Myopia	<−1.0 D [23] High myopia according to Hu et al. [7]							
	Age (years)	<9		9	10	11	12	13
	Myopia (D)	−1		−2	−2.75	−3.5	−4	−4.5
	Future potential myopes: sphere smaller than the minimum expected for their age [24]							
	Age (years)	<8	8	9	10	11	12	13
	Min (D)	0.75	0.625	0.5	0.38	0.43	0.46	0.5
Hyperopia	Greater than expected spherical equivalent for age [24]							
	Age (years)	<8	8	9	10	11	12	13
	Min (D)	1.37	1.25	1.1	1	1.03	1.09	1.12
Astigmatism	Cylinder > 1.5 D							
Amblyopia Risk	Risk factors according to American Association for Pediatric Ophthalmology and Strabismus (AAOPS) [25]: Anisometropia > 1.5 D Hyperopia > 3.5 D Myopia < −3.0 D Astigmatism > 1.5 D @ 90 ± 10° or @ 180 ± 10° Oblique astigmatism > 1.0 D							

## Data Availability

Data are available online.

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
