# Peer review of "Early Detection of Refractive Errors by Photorefraction at School Age"

_ijerph, 2022, doi:10.3390/ijerph192315880_

Round 1

Reviewer 1 Report

This paper is extremely important in identifying the rate of myopia among children in Spain. Please consider the following points.

1.         Were there any non-measurable cases? Please indicate the percentage and the cause.

2.         This study confirms the reproducibility of the results by two consecutive measurements. However, since the results do not exist, please provide the results.

3.         After confirming reproducibility, is there any such thing as the greater the refractive error, the lower the reproducibility?

4.         It is stated that after two consecutive measurements, if the values deviated from each other, the measurements were taken again. If there are two measurements, which data is adopted and analyzed? Also, are there any criteria for re-measuring after two consecutive measurements? If there are no criteria, observer bias may exist.

5.         Are most Spanish elementary school students white?

6.         Please indicate the results for gender differences.

Author Response

We really appreciate the reviewer’s opinion about the potential impact of the manuscript. In the attached file, we include a point-by-point answer to all the questions of the reviewer, and we thank all suggestions, which have been improved the manuscript.

The file with the raw data of the measurements have been sent to the editor.

Reviewer 2 Report

Early detection and treatment of refractive defects during school age are essential to avoid irreversible future vision loss and potential school problems. This study described for the first time a large cohort of children screened for refractive problems with photorefraction in Europe. The authors also presented for the first time the model for risk of high myopia development in white population in Europe. However I consider this MS to be of interest of our readers it needs a major revision before further consideration of publication process. The authors should add to the introduction and discussion sections the information about the cost effectiveness of this method comparing to others (there is some literature about that). And also the information about future managament of those afected children (are they under observation with the same method ?). The authors should also discuss the prevalence of refarctive errors in the studied population with other studies from Europe. The last remark is to improve the conclusions (the information that betwen 20-40 % of children had refractive errors is to broad).

Author Response

Authors would like to thank the reviewer that he highlighted the novelty of our study, in which we present a photorefraction screening in a large population in Europe and the prediction of risk of suffer high myopia and amblyopia in a white population. The suggestions of the reviewer have helped us to improve specially introduction and discussion of the manuscript. 

Round 2

Reviewer 1 Report

Thank you very much for submitting your revisions.

This paper has been accurately revised and is considered worthy of publication in this journal.

Reviewer 2 Report

I am staisfied with authors' reply